# Hemolytic Activity, Cytotoxicity, and Antimicrobial Effects of Human Albumin- and Polysorbate-80-Coated Silver Nanoparticles

**DOI:** 10.3390/nano11061484

**Published:** 2021-06-03

**Authors:** Dmitry Korolev, Michael Shumilo, Galina Shulmeyster, Alexander Krutikov, Alexey Golovkin, Alexander Mishanin, Andrew Gorshkov, Anna Spiridonova, Anna Domorad, Alexander Krasichkov, Michael Galagudza

**Affiliations:** 1Almazov National Research Centre, 197341 Saint-Petersburg, Russia; dimon@cardioprotect.spb.ru (D.K.); shulmeyster_ga@almazovcentre.ru (G.S.); ankrutikov@yandex.ru (A.K.); golovkin_as@almazovcentre.ru (A.G.); mishaninssma@yandex.ru (A.M.); galagudza_mm@almazovcentre.ru (M.G.); 2Pavlov First Saint-Petersburg State Medical University, 197022 Saint-Petersburg, Russia; annaasbac@mail.ru (A.S.); domorad@yandex.ru (A.D.); 3Saint-Petersburg Electrotechnical University “LETI-ETU” Named after V.I.Ulyanov (Lenin), 197376 Saint-Petersburg, Russia; krass33@mail.ru; 4Saint-Petersburg Research Institute of Influenza, 197376 Saint-Petersburg, Russia; angorsh@yahoo.com; 5Federal State Institution Saint-Petersburg Pasteur Research Institute of Epidemiology and Microbiology, 197101 Saint-Petersburg, Russia

**Keywords:** silver nanoparticles, chemical synthesis, hemolysis, cytotoxicity, antimicrobial activity

## Abstract

In this study, we aimed to develop a technique for colloidal silver nanoparticle (AgNP) modification in order to increase their stability in aqueous suspensions. For this purpose, 40-nm spherical AgNPs were modified by the addition of either human albumin or Tween-80 (Polysorbate-80). After detailed characterization of their physicochemical properties, the hemolytic activity of the nonmodified and modified AgNPs was investigated, as well as their cytotoxicity and antimicrobial effects. Both albumin- and Tween-80-coated AgNPs demonstrated excellent stability in 0.9% sodium chloride solution (>12 months) compared to nonmodified AgNPs, characterized by their rapid precipitation. Hemolytic activity of nonmodified and albumin-coated AgNPs was found to be minimal, while Tween-80-modified AgNPs produced significant hemolysis after 1, 2, and 24 h of incubation. In addition, both native and Tween-80-covered AgNPs showed dose-dependent cytotoxic effects on human adipose-tissue-derived mesenchymal stem cells. The albumin-coated AgNPs showed minimal cytotoxicity. The antimicrobial effects of native and albumin-coated AgNPs against *S. aureus*, *K. pneumonia*, *P. aeruginosa*, *Corynebacterium* spp., and *Acinetobacter* spp. were statistically significant. We conclude that albumin coating of AgNPs significantly contributes to improve stability, reduce cytotoxicity, and confers potent antimicrobial action.

## 1. Introduction

The growing incidence of antimicrobial drug resistance in a number of bacterial species is becoming a substantial barrier for the effective treatment of many infectious diseases. More than 2 million individuals are affected by nosocomial infections annually in the United States, and approximately 70% of such infections are caused by multiple antibiotic-resistant strains of bacteria [1,2]. The infections caused by these types of pathogens are associated with higher morbidity and mortality, increased risk of adverse outcomes after medical interventions, increased treatment costs, and prolonged hospital stay [3]. Altogether, these factors result in a huge economic burden on the healthcare systems. As a result, antimicrobial resistance is recognized by the World Health Organization as one of the main hindrances to the improvement of the global health, and thus warrants the immediate action of the scientific community to overcome this problem [4]. In principle, antimicrobial drug resistance can develop in any bacteria, but it has been most commonly registered in the group of bacteria known as ESKAPE, which comprises *Enterococcus faecium*, *Staphylococcus aureus*, *Klebsiella pneumoniae*, *Acinetobacter baumannii*, *Pseudomonas aeruginosa*, and *Enterobacter* spp. [5]. 

Along with the use of new antibiotics, bacteriophages, and antimicrobial peptides, the problem of antimicrobial drug resistance could be solved by the utilization of nanoparticles with antimicrobial properties [6]. In this regard, silver nanoparticles (AgNPs) have been the most intensively studied nanoparticles during the last decade owing to their unique bactericidal effect, which far exceeds that of silver ions [7]. The mechanisms involved in AgNPs antimicrobial activity are under intense investigation and may include permeabilization of bacterial cell wall/membrane, stimulation of reactive oxygen species generation, and interaction of AgNP-derived silver ions with DNA with subsequent disruption of DNA transcription [8]. Furthermore, the magnitude of AgNPs’ biocidal effect depends on several characteristics, such as their shape and size, coating type, and the duration of contact between the nanoparticles and bacterial cells. 

Undoubtedly, AgNPs’ marked bactericidal activity against antibiotic-resistant microorganisms represents their major advantage for application. However, it is also important to analyze the toxicological profile of AgNPs—that is, their safety for eukaryotic cells, both in vitro and in vivo. Ideally, any nanoparticle-based antimicrobial agent should exhibit potent microbicidal activity while being nonhazardous to healthy mammalian cells. The issues on AgNP safety have recently gained significant attention, and the studies performed have yielded controversial data. For instance, AgNPs have been shown to exert toxic effects on rat hepatocytes and neurons in vitro [9], murine stem cells, and human lung epithelial cells [10]. Vuković B. reported that bovine serum albumin-, poly(vinylpyrrolidone)-, poly-L-lysine-, and bis(2-ethylhexyl)-sulfosuccinate-sodium-coated AgNPs show dose-dependent cytotoxicity against human mononuclear cells [11]. In this study, the most prominent cytotoxic properties were found in positively charged poly L-lysine- and bovine-serum-albumin-coated AgNPs. Biologically synthesized bare AgNPs were three times more toxic to HEK-293 cells than HeLa cells [12]. Based on these and other studies, it seems reasonable to conclude that, in general, the toxicological properties of AgNPs depend on their size, shape, type of coating, zeta potential, and the biological test system used. One of the central problems with using unmodified AgNPs is their poor stability in aqueous media, which results in the formation of aggregates and loss of the unique properties that are typical of nonagglomerated nanoparticles. In order to circumvent this issue, several coating materials have been suggested for AgNPs, including bovine serum albumin, chitosan [13], polyethyleneimine [14], polyvinylpyrrolidone [15], polyethylene glycol [16], etc. In general, the use of different coatings was associated with improved AgNP stability in water. In certain cases, nanoparticle coating resulted in augmented antibacterial activity compared with bare nanoparticles; nonetheless, the extent of this effect seems to be coating-dependent. For example, Izak-Nau et al. showed that polydopamine-covered AgNPs present a more pronounced antibacterial effect against *E. coli*, which was associated with a more intensive reactive oxygen species generation than poly(vinylpyrrolidone)-coated AgNPs. The chemical composition of the coating could affect not only the intensity of the antimicrobial effect, but also the toxicological properties of the nanoparticles. Decoration of AgNPs with N-acylated homoserine lactonase protein resulted in enhanced bactericidal activity against multidrug-resistant *K. pneumoniae* and reduced organ toxicity in mice compared to native AgNPs [17]. At present, no information is available on the modulation of antimicrobial and cytotoxic effects of AgNPs after coating with human albumin and polysorbate-80.

Therefore, in this study, we aimed to synthesize and characterize albumin- and polysorbate-80-coated AgNPs using transmission electron microscopy, dynamic light scattering, absorption, and X-ray spectrometry. Next, we studied the hemolytic activity of AgNPs with two different coatings on human red blood cells, as well as their cytotoxicity in human adipose-tissue-derived mesenchymal stem cells (MCSs). After confirming minimal cytotoxicity of albumin-coated AgNPs, we studied their antimicrobial activity against some of the bacterial species belonging to the ESKAPE group—in particular, *S. aureus*, *K. pneumonia*, *P. aeruginosa*, *Corynebacterium* spp., and *Acinetobacter* spp.—and compare it to that of uncovered AgNPs.

## 2. Materials and Methods

### 2.1. Materials and Reagents

All chemicals used were of analytical grade and purchased from Sigma-Aldrich (St. Louis, MO, USA), unless otherwise specified.

### 2.2. Preparation of AgNPs

AgNPs were synthesized by chemical reduction of silver nitrate in an aqueous phase using sodium citrate as reducing agent [18]. Briefly, a mixture of 6.25 mL of water, 1.25 mL of sodium citrate (1% by weight), 1.25 mL of silver nitrate (AgNO_3_; Product Number 209139, Sigma-Aldrich, St. Louis, MO, USA) (1% by weight), and 50 μL of potassium iodide (300 μM) were prepared by stirring at room temperature (25 ± 1 °C) and kept at that temperature for 5 min. The mixture was poured into 237.5 mL of boiling water, which included 250 μL of ascorbic acid (0.1 M), and stirred. The color of the solution changed to yellow and then slightly to orange, indicating the formation of nanoparticles. The colloidal solution was then boiled for 15 min. After cooling, the AgNPs were stored in a dark, glass container at 4 °C in the dark. The resulting colloidal solution was purified by dialysis using a 35-kDa membrane against distilled water.

### 2.3. Modification of Silver Nanoparticles

Isotonic solutions are used in medical drugs, particularly in saline solution (0.9% NaCl). Colloidal solutions of native AgNPs in saline (AgNPs–SS) are characterized by instability and rapid aggregation. To minimize this effect, the nanoparticles were modified by two methods to be coated with human albumin (human albumin 20%, Baxter, A-1221, Vienna, Austria) or polysorbate-80 (Tween-80, Chimmed, Moscow, Russia. Albumin-coated AgNPs (AgNPs–Alb) were prepared as follows: a AgNPs suspension (5 mL) freshly prepared in distilled water was mixed with 1 mL of 20% aqueous solution of albumin and stirred for 2 h at 300 min^−1^ speed in an LS-220 orbital shaker (LOIP, Russia). It was assumed that albumin chemical binding to the nanoparticle surface took place, which further contributed to the colloidal stability of the solution upon 0.9% NaCl addition. 

To coat the AgNPs with Polysorbate-80, 50 μL of Tween-80 was added to 5 mL of AgNP solution and stirred for 2 h at 300 min^−1^ in an LS-220 orbital shaker. It was assumed that the surfactant covered the surface of the (AgNPs–Tween) nanoparticles, which further contributed to the colloidal stability of the solution upon the addition of 0.9% NaCl. 

The biological studies were carried out on the three types of nanoparticles: AgNPs–SS, AgNPs–Alb, and AgNPs–Tween. Taking into account the instability of the native AgNPs, they were prepared immediately before the tests by mixing the AgNPs with 0.9% NaCl.

### 2.4. Characterization of AgNPs

AgNP concentration in the resulting solutions was determined by evaporation on a watch glass [18] at 100 °C using a a Zetasizer Ultra (Malvern Instruments Ltd., Great Britain, Malvern, UK). Briefly, the AgNP suspensions were placed on the watch glass and covered with filter paper to prevent contamination, and then dried. To maximize drying speed, the watch glass was placed in a thermostat, ensuring good air circulation for 24 h. The AgNPs’ dry powder weight was determined. 

Size distribution by volume, zeta potential distribution, polydispersity index (PDI), and cumulative particle concentration were determined using a Zetasizer Ultra (Malvern Instruments Ltd., Malvern, UK).

X-ray spectra of AgNPs were obtained with an energy dispersive X-ray fluorescence spectrometer, EDX 800 HS series (Shimadzu, Japan).

The resulting concentrate of the nanoparticle solution was filtered through Vivaspin 6 membrane filters 1000 kDa, with pore sizes of 0.2 μm (Sartorius, Germany) and PES membrane material.

Transmission electron microscopy (TEM) images were obtained using a JEM-2010 instrument (JEOL, Tokyo, Japan).

Ultraviolet-visible (UV-Vis) absorption spectroscopy analysis was performed on the AgNP samples dispersed in water using a UV-Vis spectrophotometer (Unico 2802s, Unico Sys, Franksville, WI, USA).

### 2.5. Hemolytic Activity of AgNPs

The hemolytic activity [19] of bare and coated AgNPs was studied in whole blood of two healthy donors (blood of donor #1, OI Rh- k- dee; blood of donor #2, AB IV Rh+ k- DCCee). Then, 0.5 mL of each of the three AgNP tested samples were separetely incubated with an equal volume of blood from each of the donors for 24 h in a biological thermostat at 37 °C. Blank samples were prepared; distilled water was added as a positive control (PC) and saline as a negative control (NC). After incubation, the samples were centrifuged for 20 min at 2000 min^−1^.

Hemolytic activity was assessed using the hemolysis coefficient, which was determined spectrophotometrically (Unico 2802, Unico Sys, USA) based on the optical density of the samples at a wavelength of 415 nm, corresponding to the absorption band of oxyhemoglobin. To measure optical density, 200 μL of the sample was brought up to 6 mL with saline solution.

Hemolytic activity was calculated using the following formula:*HC* = (*O* − *NC*)/(*PC* − *NC*) × 100%,
where *O* is the measured optical density of the sample, *NC* is the negative control (0% hemolysis of the blank sample), and *PC* is the positive control (100% hemolysis of the blank sample).

### 2.6. Cytotoxic Properties of AgNPs

We studied the cytotoxic properties of AgNPs–SS, AgNPs–Alb, and AgNPs–Tween at concentrations of 1 and 10%. The experiment was carried out on 12 mm-diameter cover slips in duplicates. Human multipotent mesenchymal stem cells (MSCs) obtained from the subcutaneous adipose tissue of healthy donors were used. The study was performed according to the Helsinki declaration and approval was obtained from the local Ethics Committee of the Almazov National medical research centre (№ 12.26/2014; 1 December 2014). Written informed consent was obtained from all subjects prior to fat tissue biopsy. MSCs were cultivated in alpha-MEM supplemented with 10% fetal bovine serum (FBS), 1% L-glutamine, and 1% penicillin/streptomycin solution and incubated at 37 °C with 5% CO_2_ content.

A total of 23 sterile coverslips were placed in the wells of a 24-well plate (10 groups of 2 coverslips, and 3 coverslips in the control group). To each well, 1 mL of MSC suspension at a concentration of 50,000 cells/mL was added and cultured for 24 h. Then, 10 μL (1%) and 100 μL (10%) of the AgNP preparations and phosphate-buffered saline (PBS) (as control) were added, and the cells were incubated for 48 h at 37 °C with 5% CO_2_ content. The AgNP preparations and PBS were preheated in a water bath. A previously prepared sample of sodium chloride was dissolved in the aqueous solution of native AgNPs immediately before its addition to the plate. After incubation, the coverslips were washed with PBS to remove remains of the medium and were then fixed with 4% paraformaldehyde (PFA) solution for 10 min.

After fixation, the cover slips were washed with PBS and subjected to immunocytochemical staining with antibodies against the cytoskeleton protein vinculin. Briefly, the coverslips with the attached cells were treated with 0.05% Triton X-100 solution for 3 min, followed by washing with PBS. Then, to reduce nonspecific binding of antibodies, the coverslips were blocked with a PBS 1% fetal bovine serum (FBS) for 30 min. After incubation, the FBS solution was removed and human antivinculin primary mouse monoclonal antibody (1:200 in 1% FBS–PBS) was added to the wells, which were incubated for 1 h at room temperature. The coverslips were washed in PBS and then incubated with secondary goat antimouse IgG antibody labeled with Alexa Fluor 568 fluorochrome (1:1000 in 1% FBS-PBS) for 1 h at room temperature. The coverslips were thoroughly washed with PBS to remove the antibody residues.

Finally, cell nuclei were stained with DAPI (4,6-diamidino-2-phenylindole). For this purpose, the DAPI stock solution was diluted in PBS (1:40,000), then added to the wells and incubated for 40 s, after which the coverslips were thoroughly washed with PBS to eliminate the dye residues. After staining, the coverslips were mounted on object-plates (slides) using a mounting medium and stored in the dark at room temperature. The MSCs were quantitatively and qualitatively analyzed by fluorescence microscopy. Cells were visualized using an Axiovert inverted fluorescence microscope (Zeiss, Germany) and a compatible Canon camera. DAPI fluorescence was recorded using an appropriate filter, and Alexa Fluor 568 fluorescence was recorded using the rhodamine channel. Quantitative analysis of cells was carried out by counting the stained cell nuclei in 10 fields of view (×10) for each technical repetition, followed by recalculation of the number of MSCs per mm^2^. Dose-dependent cytotoxicity was assessed using two concentrations of each AgNP: 1% and 10%. For the qualitative analysis, cell morphology was assessed based on the stained cytoskeleton in 10 fields of view (×40) for each technical repetition.

Cytotoxicity was assessed by AgNP and MSC cocultivation for 3 days. The cells located on the surface of the culture were used as controls. All the samples, including the controls, were analyzed in triplicates. Then, the cells were removed from the AgNP samples with trypsin 1% EDTA solution, resuspended in annexin binding buffer (Biolegend), and stained with annexin V FITC (BioLegend) and propidium iodide (Sigma Aldrich) for 20 min in the dark, according to the manufacturer’s recommendations. The samples were analyzed by flow cytometry in a Guava EasyCyte8 flow cytometry laser (Millipore). The data obtained were analyzed using the Kaluza 2.1 software (Beckman Coulter). Double positive events were identified as cells in late apoptosis or necrosis state; Annexin V positive corresponded to cells in early apoptosis; and double negative to viable cells [20,21,22].

### 2.7. Antibacterial Assay

The AgNPs–Alb samples were prepared under aseptic conditions as follows: the samples were placed in 0.2 μm centrifugal filters (Sartorius, Germany) with a PES membrane material and centrifuged for 5 min at 4200 min^−1^ speed. Then, in a laminar flow cabinet, the resulting centrifugate was poured into sterile tubes, sealed, and sent for analysis. The antimicrobial effect of samples was assessed using the disk-diffusion agar method test [23].

The antimicrobial effect of the preparations was studied on the collection strains *S. aureus* ATCC 25923, *K. pneumonia* ATCC 70060, *P. aeruginosa* ATCC 27853, and clinical isolates, *Corynebacterium* spp. #4492, and *Acinetobacter* spp. #6905.

The inocula were prepared as follows: The pure microbial cultures were subcultured (36 ± 1 °C, 18–20 h). A suspension of 1.5 × 10^8^ CFU/mL was prepared from a one-day-old culture according to the McFarland 0.5 turbidity standard. The optical density of the microbial suspension was monitored by densitometry (Densimat, BioMerieux, USA, France). The suspension was applied to the surface of petri dishes containing Mueller-Hinton agar (BioMedia, Russia), and 8-mm wells were made in the agar. A total 100 μL of AgNPs–Alb sample was added to the experimental wells (two per dish), and 100 μL of sterile 0.9% NaCl solution was added to the control wells (one per dish). The plates were incubated at 36 ± 1 °C for 18–20 h. The growth inhibition zones around the wells were measured. The experiment was repeated three times and the average value was recorded.

### 2.8. Statistical Analysis

Statistical processing of the obtained data on AgNP cytotoxicity was performed in the GraphPad Prism 9 software using the Mann–Whitney nonparametric U-test. Results were presented as mean ± standard deviation (SD).

## 3. Results

### 3.1. Physicochemical Properties of AgNPs

The concentration of the synthesized colloidal solution of silver nanoparticles purified from impurities was 0.003 wt%.

The number of particles determined by the dynamic light scattering (DLS) method was 5.503 × 10^15^ particles/mL (Figure 1a), which corresponds to 0.0015 g/mL.

Size distribution by volume has two peaks. The first one is a narrow high peak (size, 4.6 nm; volume, 23.3%), and the second is very small and almost flat (size, 46 nm; volume, 0.043%) (Figure 1b). Moreover, the first peak is much larger than the second, indicating the homogeneity of the fractional composition of NPs, and the PDI = 0.55 of AgNPs confirms this characteristic.

The average value of AgNPs’ zeta potential is −46.56 mV. This large negative charge indicates that the particles were stable in solution.

In addition, albumin-coated and polysorbate-80-coated silver nanoparticles data are presented together in Figure 1, for better comparison of their properties. 

Albumin-coated silver nanoparticles have a concentration of 4.16 × 10^14^ particles/mL; its size distribution by volume presents two peaks. The first peak was narrow and high (size, 7.14 nm; volume, 22.8%), and the second was small (size, 32.3 nm; volume, 0.57%). The average value of the zeta potential of albumin-coated silver nanoparticles was −14.63 mV. Polysorbate-80-coated silver nanoparticles have a concentration of 4.05 × 10^14^ particles/mL; its size distribution by volume has one peak, narrow and high (size, 5.28 nm; volume, 22.65%). The average zeta potential of polysorbate-80-coated silver nanoparticles was −27 mV.

The X-ray spectra of AgNPs (Figure 2) show that there was 100% Ag in the lyophilized suspension sample.

Membrane mediated filtration showed that 100% of the nanoparticles passed through membranes with pore sizes of 1000 kDa and 200 nm. Thus, the colloidal solution is stable over time and maintains its properties for a long time period (12 months).

TEM images (Figure 3a) showed that the colloidal solution contained spherical silver nanoparticles of 40 nm in size. This size corresponded to the largest fraction determined by DLS. The particles were completely electron-dense; thus, their structure could not be revealed.

The study of the absorption spectra of the silver nanoparticles colloidal solution (Figure 4) showed that the absorption bands corresponding to the plasmon effect (maximum 434 nm) correlated with those previously described in the literature [24]. Albumin-coated silver nanoparticles and polysorbate-80-coated silver nanoparticles saved the same peaks of maximum (Figure 4) after the particles had been coated, which indicates size preservation. The slight decrease in intensity after passing the 1000-kDa membrane filter was probably associated with silver nanoparticle adhesion to the filter material.

### 3.2. Hemolytic Activity of AgNPs Preparations

The study of the hemolytic activity of AgNP samples (Table 1) did not reveal a negative effect exerted by AgNPs–SS and AgNPs–Alb on the blood obtained from healthy donors. The hemolysis coefficient of these samples was found to be indistinguishable from the background level and did not exceed 1%. Moreover, despite the fact that Tween-80 is a component of several vaccines, its potent hemolytic activity was registered in AgNP–Tween samples. The value of the hemolysis coefficient for this sample after one hour exceeded 1%, and after 24 h was above 30%. 

### 3.3. Cytotoxicity of AgNPs Preparations

When calculating the number of cells on the coverslip surface (Table 2), no significant differences were observed between the number of adhered cells in the control group, the groups with added PBS (1% and 10%), or the AgNPs–SS and AgNPs–Alb preparations at a concentration of 1% (*p* > 0.05, Mann–Whitney test).

Qualitative analysis of cell morphology showed that the cells formed a confluent monolayer on all the coverslips, independently of the group (Figure 5); were well spread on the glass surface; and had a typical elongated shape with pronounced cytoplasm and diffusely stained protein vinculin. In the AgNPs–Tween and AgNPs–Alb groups, stiff structures (foci of local adhesion) were clearly visualized. In addition, in the AgNPs–Tween groups, both at 1% and 10%, the MSCs were located less densely; in the AgNPs–Tween group at 10% concentration, the MSC cytoplasm presented a honeycomb structure.

When AgNPs–Tween was added to the cultures at a volume of 100 μL, the cells were completely damaged, and in the course of laser flow cytometry, were later not identified as integral objects, but only as cell debris (Figure 6, Table 3). Furthermore, AgNPs–Tween addition, even at a dose of 10 μL, resulted in a significant decrease in the proportion of living cells due to an increase of cells in the early apoptotic stage. A similar pattern was observed with a higher dose.

Despite the stable level of living cells after AgNPs–Alb addition, apoptosis increased with the introduction of the minimum dose. Changes similar to those of the AgNPs–Tween preparation were observed.

### 3.4. Antimicrobial Effects of AgNPs

Since the AgNPs–Tween preparation showed the highest extent of cytotoxicity, its antimicrobial activity was not tested. Both AgNPs–SS and AgNPs–Alb preparations possessed antimicrobial effects (Figure 7, Table 4) against all the tested cultures—that is, collection strains *S. aureus*, *K. pneumonia*, and *P*. *aeruginosa*, and clinical isolates *Corynebacterium* spp. and *Acinetobacter* spp. Growth inhibition zones ranged from 10 to 18 mm (Figure 7). In the collection strains, the largest zone of growth inhibition was observed in *K. pneumoniae* cultures, amounting to 16 ± 1 mm for AgNPs–SS, and 14 ± 1 mm for AgNPs–Alb. When tested on clinical isolates, this value was the highest in the *Corynebacterium* spp. culture, showing values of 16 ± 1 mm and 18 ± 1 mm for AgNPs–SS and AgNPs–Alb, respectively. No statistically significant differences were found between the antimicrobial activities of AgNPs–SS and AgNPs–Alb.

## 4. Discussion

In this study, we used chemical reduction in the liquid phase for the synthesis of prototypical AgNPs. In order to increase their stability in water, two different coatings were used: human albumin and polysorbate-80 (Tween). Irrespective of the coating type, a significant increase in AgNP stability was observed; this was followed by a detailed analysis of the physicochemical parameters of both intact and modified AgNPs. Since the basic physicochemical properties of AgNPs were not altered by the presence of either coating, we tested AgNPs’ biological activity relevant to their safety profile. Unexpectedly, Tween-coated AgNPs were found to cause significant hemolysis compared with uncovered and albumin-covered AgNPs. The same result was obtained in the cytotoxicity assay using adipose-tissue-derived mesenchymal stem cells, where even a low dose of Tween-coated AgNPs elicited significant changes in cell morphology and an increased proportion of apoptotic cells. The final experimental series demonstrated that albumin-coated and bare AgNPs were equally effective in terms of bactericidal activity against pathogens of the ESKAPE group. 

At present, several methods of AgNP synthesis have been proposed, including biological, physical, and chemical methods. Biological methods are called green chemistry methods [25,26,27] because their use is associated with saving resources and minimal release of hazardous materials into the environment. Biological methods are based on nicotinamide-adenine-dinucleotide-reductase-mediated enzymatic reduction of silver salts into AgNPs. However, one limitation of this method is the relatively low yield of the product, which is higher when using plant extracts compared to the use of bacteria and fungi. In physical methods, AgNPs are synthesized by physical pulverization of metals, which excludes contamination of the product with solvents but requires a high-energy source and special equipment. The most extensive group of AgNP synthesis techniques is the chemical one. Among the chemical methods, the reduction technique is the most commonly used. In this study, we selected a chemical reduction method for AgNP production because of its inexpensiveness, technical feasibility, and reproducibility. The chemical reduction method is based on the obligatory use of two components: a silver source and a reducing agent. Many different compounds have been used as reducing agents in chemical AgNP synthesis, such as hydrogen peroxide, sodium borohydride, sodium citrate, and gallic acid, among others [28]. In this study, we used silver nitrate and sodium citrate as the silver source and reducing agent, respectively, which resulted in the generation of monodisperse AgNPs with an average diameter of 40 nm. The absorption band of the colloidal solution of AgNPs corresponds to the plasmon effect (maximum 434 nm) and correlates with that previously described in the literature [24]. 

It is well-established that AgNPs exhibit potent bactericidal activity [29]. This effect is mainly explained by the release of silver ions (Ag^+^) from their extensive surface, which interact with thiol groups of essential enzymes and proteins of prokaryotic organisms leading to the disruption of energy production and ensuing cell death. In recent studies, the putative antiviral effects of AgNPs have been discussed, although the mechanisms of AgNP-mediated virus inhibition are less clear [30]. In general, AgNPs have been used in biomedicine for provision of the effect of “silver battery”, resulting in a more stable and prolonged Ag^+^ release compared to silver salts. Nonetheless, unmodified AgNPs are characterized by very poor stability in the aqueous phase, which has stimulated researchers to develop various types of organic coatings aimed at increasing the stability of AgNPs and optimizing some other properties of these nanoparticles [31]. For instance, AgNPs coated with polyethylene glycol (PEG) were associated with a decreased polydispersity index compared with nonmodified AgNPs [32] and, at the same time, had no influence on the bactericidal activity of biologically synthesized AgNPs. Lysozyme-covered AgNPs showed a triclozan-like bactericidal effect on multiresistant strains of *K. pneumoniae* linked to inhibition of the type II fatty acid biosynthesis pathway and, additionally, to the development of Ag^+^-mediated oxidative stress in bacteria [33]. The coating of AgNPs with chitosan and bovine serum albumin has been associated with a significant increase in their stability and, in addition, with the presence of marked bactericidal effects against seven oral and nonoral bacteria, provided that the magnitude of the antimicrobial effect was higher in chitosan-covered AgNPs with smaller diameters [13]. Polydopamine coating of 3–25 nm-thickness on the surface of AgNPs ensured augmentation of bactericidal effects of AgNPs, potentially due to ionic coordination interaction between silver ions and catechol groups of polydopamine, which has been associated with increased reactive oxygen species generation [15]. In the present study, the coatings used were albumin and Tween; both of them significantly enhanced AgNPs’ stability in aqueous phase. Albumin coating did not increase the antibacterial activity of AgNPs, but at the same time, it had no inhibitory influence on this parameter compared with uncovered AgNPs.

Another critical aspect of the use of AgNPs is their safety in mammalian cells. In general, uncovered AgNPs are characterized by an appropriate safety profile, although several recent publications have shown dose-dependent cytotoxicity of AgNPs. For example, Chang et al. (2021) showed that AgNPs can cause autophagy and apoptosis in a mouse hippocampal neuronal cell line (HT22 cells) [34]. Other studies have demonstrated that biologically derived AgNPs present significantly higher cytotoxicity against normal mammalian HEK-293 cells than HeLa tumor cells [12], which questions the therapeutic effectiveness of these nanoparticles against tumors. Intratesticular administration of AgNPs to male mice resulted in pronounced, albeit reversible, disorders of spermatogenesis [35]. Despite the presence of these disturbing reports, the toxicity of unmodified AgNPs is not burgeoning, partly because the perspective of contact and topical AgNP use is clinically more feasible than the parenteral route of AgNPs administration. The problem of nanoparticle coating material safety may be regarded as a more serious issue, which has been confirmed by the present study. Ideally, the coating should provide stabilization of AgNPs and enhance its antimicrobial activity while being biologically inert and compatible. Tween 80 (or polysorbate) has been used for the stabilization of AgNPs with fairly good results and antibacterial effectiveness [36]. However, the mentioned study did not contain any data on the toxicity of Tween-coated AgNPs. While we obtained encouraging results on the stability of Tween-coated AgNPs, at the same time, we observed pronounced hemolytic activity and cytotoxicity of such nanoparticles, which was exclusively due to the properties of the coating material, as the effects of Tween-coated AgNPs were compared with those of bare nanoparticles. Tween-80 is a relatively common amphiphilic coating material that is currently used for solubilization and stabilization of various nanoparticles in aqueous media. For example, Tween- and chitosan-decorated alginate nanocapsules containing rifampicin and vitamin C have demonstrated appropriate pulmonary biocompatibility and a uniform distribution throughout the lung lobes in rats after intratracheal instillation with predominant phagocytic uptake by alveolar macrophages [37]. Moreover, recombinant erythropoietin (rEPO)-loaded and Tween-covered albumin nanoparticles enhanced the transport of rEPO into the brain in a rat model of traumatic brain injury [38]. Of note, several FDA-approved drugs contain Tween. In particular, Tween is used for solubilization of docetaxel, a drug for the treatment of malignant tumors such as breast, ovarian, non-small-cell lung, and prostate cancer, and it is present in some vaccines [39]. Although viable alternatives to Tween are currently lacking, the literature contains multiple reports on the significant toxicity of Tween, which is independent of the side effects elicited by the cytostatic itself. Examples of such clinically relevant toxic effects include sensory neuropathy, nephrotoxicity, hypersensitivity reactions, and fluid retention [39,40]. In our study, the data on pronounced Tween toxicity were confirmed by hemolysis and cytotoxicity assays. In the current study, the use of Tween-coated AgNPs was associated with dramatic changes in cell morphology, including the presence of a honeycomb structure of the cytoplasm and increased stiffness, determined by vinculin expression assessment. Apparently, the cell membrane was damaged by the Tween-coated AgNPs, which did not prevent active cell adhesion, but probably led to a reduction in division rate. On the other hand, when testing the albumin-coated AgNPs, an increase in intercellular contacts was noted, and areas of focal adhesions were visualized. In general, this indicates a favorable effect of albumin, which, in this case, is playing the role of feeder. Thus, albumin-coated AgNPs were found to be safe in our model. After this result, we studied the antimicrobial potential of albumin-coated AgNPs versus native AgNPs. For this purpose, we selected pathogens belonging to the ESKAPE group, which are responsible for the majority of the reported hospital-acquired infections worldwide [41]. Originally, AgNPs were positioned as universal antimicrobial agents against a wide spectrum of pathogens and infections. However, it seems that the most feasible clinical scenario for their application is as topical treatments for skin and mucous membrane lesions of the gastrointestinal, respiratory, and urinary systems. In this study, we focused on the potential future application of AgNPs for the treatment of diabetic foot syndrome, particularly in the form of chronic ulcers and infected lesions. After analyzing the literature on the relative prevalence of certain infectious agents in this particular clinical setting, we selected a battery consisting of five microorganisms [42,43]. As a result, albumin-coated AgNPs demonstrated antibacterial activity comparable to that of native AgNPs, which is consistent with earlier studies performed on different bacterial strains [13]. Growth inhibition zones of *S. aureus*, *Acinetobacter* spp., and *P. aeruginosa* were found to be smaller than those of *Corynebacterium* spp. and *K. pneumonia*. In this sense, the diameter of the growth inhibition zone depends on several factors, including the sensitivity of a particular microorganism to a biocidal agent, biocidal drug concentration in the test system (minimum inhibitory concentration), and the rate of drug diffusion. It is difficult to determine the minimum inhibitory concentration of the test system used in the present study. The results indicate that the concentration of AgNPs used might have been insufficient to cause intensive growth inhibition of some microorganisms. It is also conceivable that the exposure time was insufficient. Nonetheless, it is evident that the use of AgNPs as antimicrobial particles is a promising tool for the treatment of infections. 

## 5. Conclusions

In conclusion, 40-nm AgNPs were synthesized by chemical reduction, followed by coating with albumin or Tween-80. Hemolytic activity of nonmodified and albumin-coated AgNPs was found to be minimal, while Tween-80-modified AgNPs produced significant hemolysis after 1, 2, and 24 h of incubation. In addition, both native and Tween-80-covered AgNPs showed dose-dependent cytotoxic effects on human adipose-tissue-derived mesenchymal stem cells. Albumin-coated AgNPs showed minimal cytotoxicity. The antimicrobial effects of native and albumin-coated AgNPs against *S. aureus, K. pneumonia, P. aeruginosa, Corynebacterium* spp., and *Acinetobacter* spp. were significantly different. We conclude that albumin coating of AgNPs results in significant improvement in their stability, reduction of cytotoxicity, and the presence of potent antimicrobial activity.

## Figures and Tables

**Figure 1 nanomaterials-11-01484-f001:**
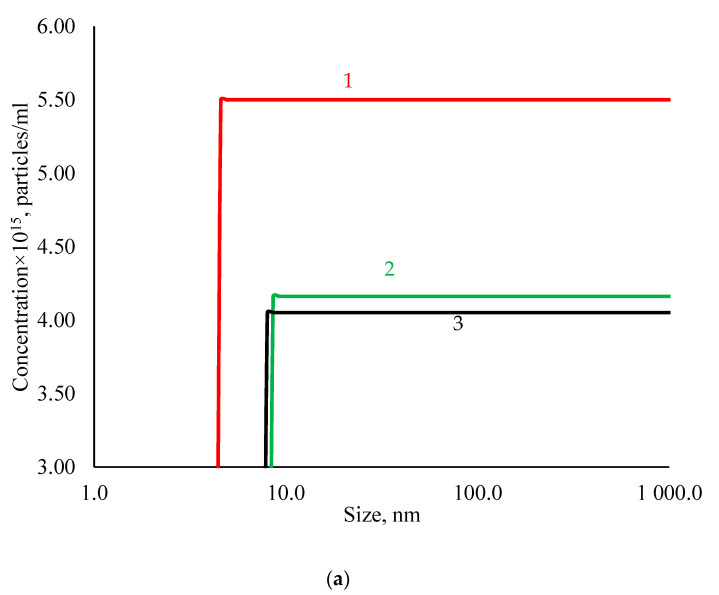
Physicochemical characteristics of AgNPs (1, initial suspension; 2, albumin-coated silver nanoparticles; 3, polysorbate-80-coated silver nanoparticles). (**a**) Cumulative particle concentration. (**b**) Size distribution of AgNPs by volume. (**c**) Zeta potential distribution.

**Figure 2 nanomaterials-11-01484-f002:**
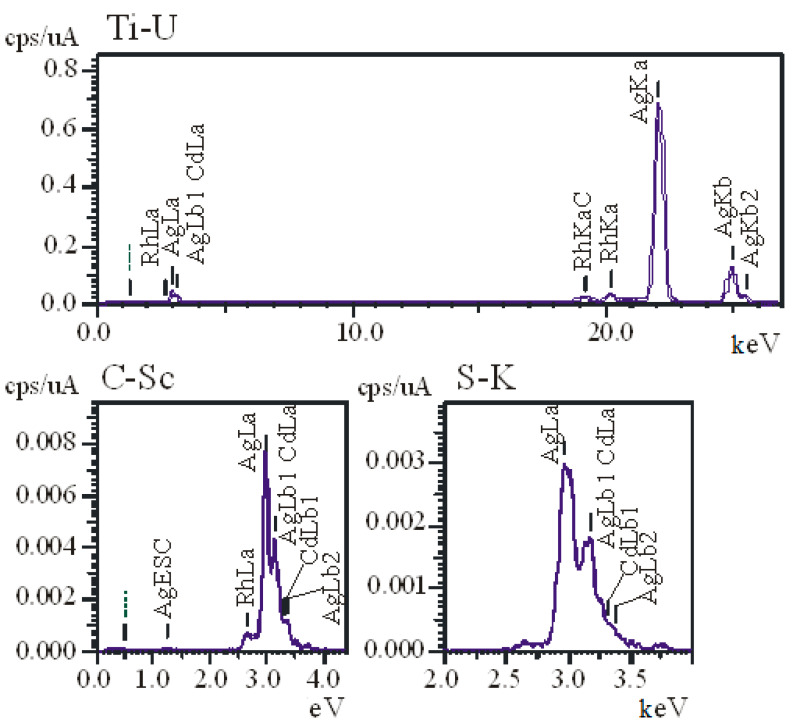
X-ray spectra of AgNPs.

**Figure 3 nanomaterials-11-01484-f003:**
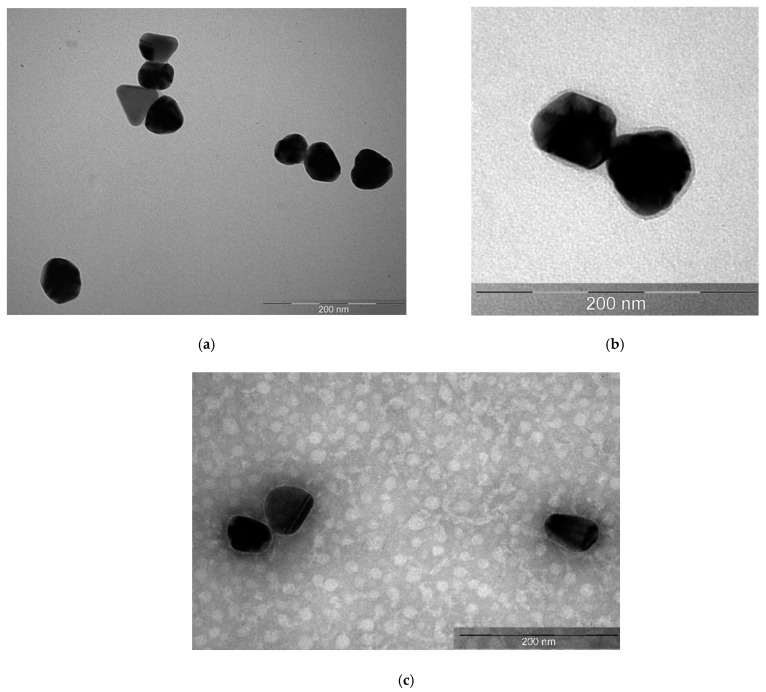
TEM photographs of (**a**) silver nanoparticles, (**b**) polysorbate-80-coated silver nanoparticles, and (**c**) albumin-coated silver nanoparticles.

**Figure 4 nanomaterials-11-01484-f004:**
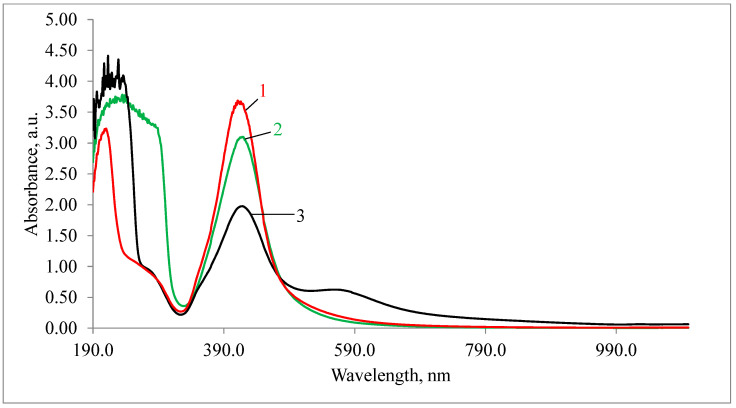
Absorption spectra of a silver nanoparticles suspension: 1, initial suspension; 2, albumin-coated silver nanoparticles; 3, polysorbate-80-coated silver nanoparticles.

**Figure 5 nanomaterials-11-01484-f005:**
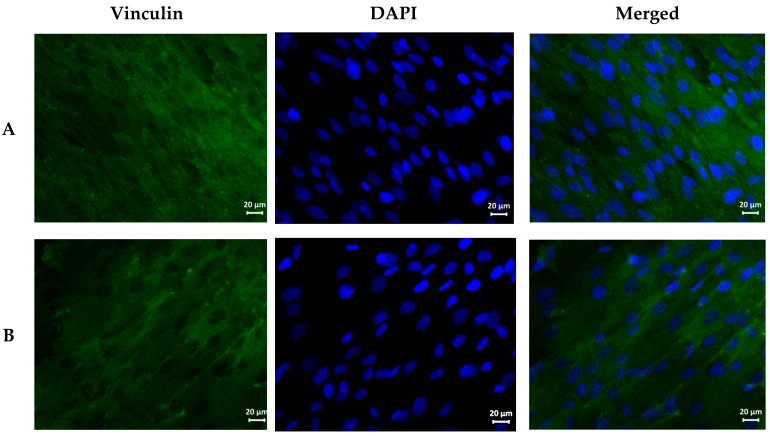
MSCs after cocultivation with the AgNP preparations. Immunofluorescence staining of vinculin, DAPI nuclei staining, X40 magnification. (**A**) Control (MSC culture medium); (**B**) PBS 10 μL; (**C**) PBS 100 μL; (**D**) AgNPs–SS 10 μL; (**E**) AgNPs–SS 100 μL; (**F**) AgNPs–Tween 10 μL; (**G**) AgNPs–Tween 100 μL; (**H**) AgNPs–Alb 10 μL; (**I**) AgNPs–Alb 100 μL.

**Figure 6 nanomaterials-11-01484-f006:**
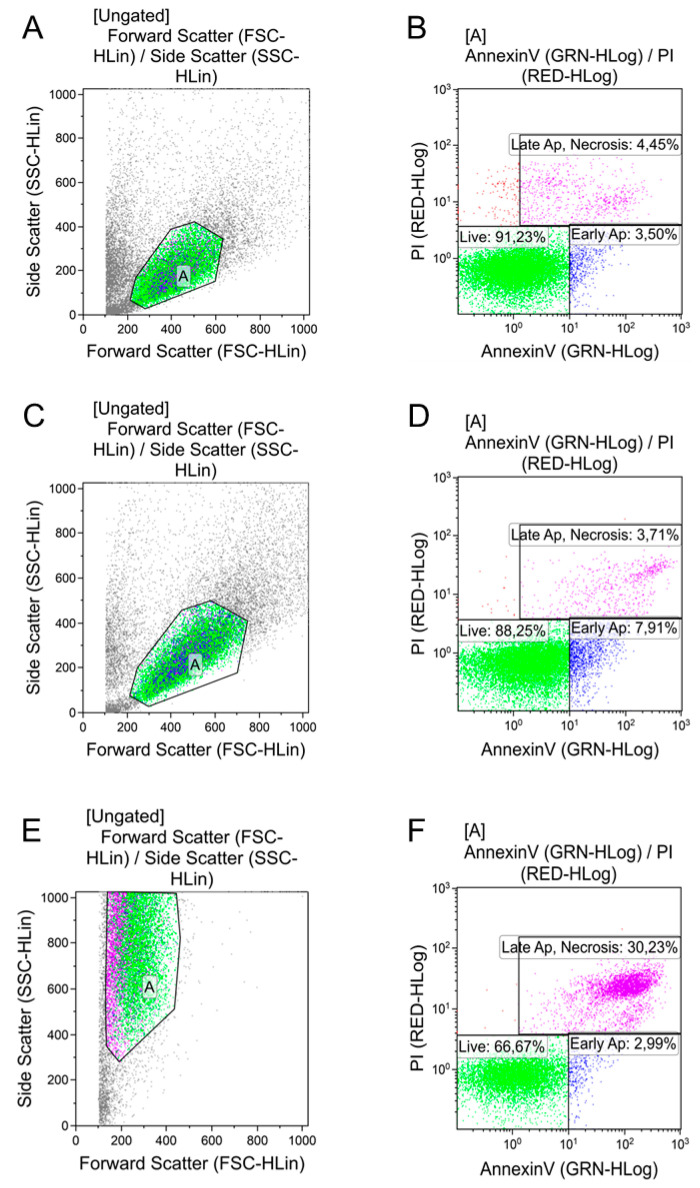
Morphology and apoptosis rates of human adipose-tissue-derived mesenchymal stem cells after incubation with PBS, albumin-, and Tween-80-coated AgNPs. Cell morphology according to forward and side scattering profiles obtained with flow cytometry analysis in (**A**) PBS-treated cells, (**C**) AgNPs–Alb 100 μL, and (**E**) AgNPs–Tween 100 μL groups. Representative scatter plots of propidium iodide (*y*-axis) versus annexin V (*x*-axis) in (**B**) PBS-treated cells, (**D**) AgNPs–Alb 100 μL, and (**F**) AgNPs–Tween 100 μL groups.

**Figure 7 nanomaterials-11-01484-f007:**
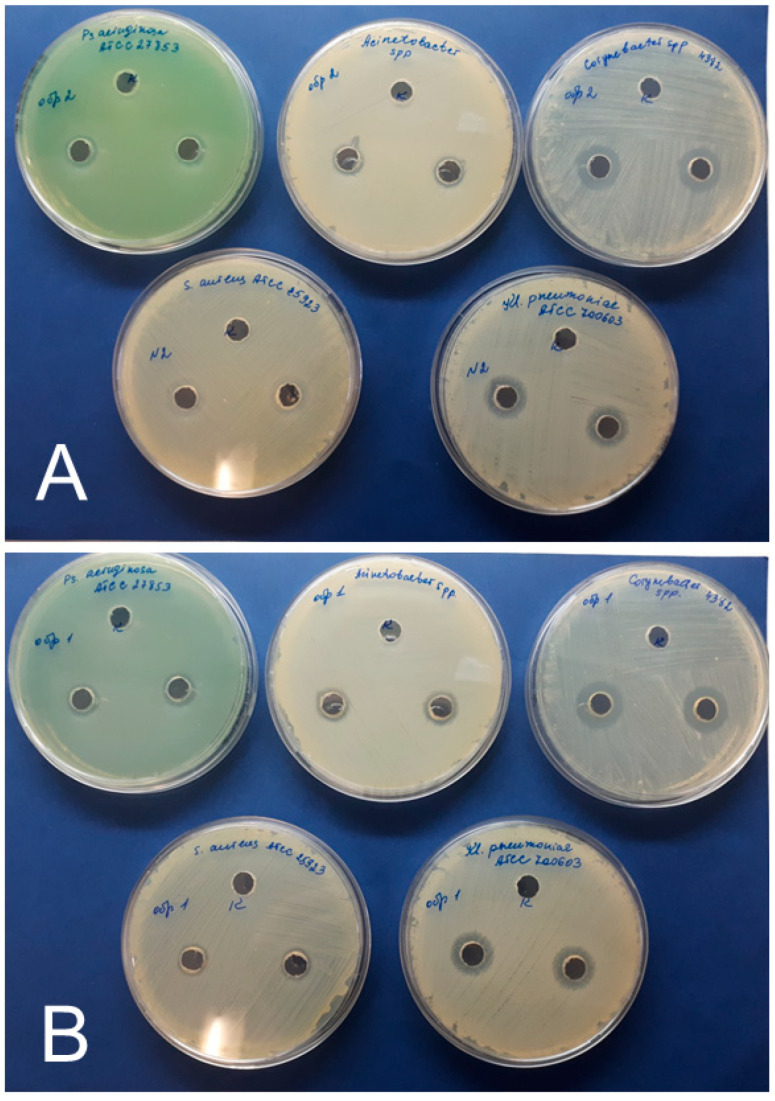
Representative figures of the inhibition zone test of (**A**) AgNPs–SS and (**B**) AgNPs–Alb using *S. aureus*, *K. pneumonia*, *P*. *aeruginosa*, *Corynebacterium* spp., and *Acinetobacter* spp.

**Table 1 nanomaterials-11-01484-t001:** Hemolysis coefficients.

Sample	Hemolysis Coefficient, %	
	Blood of Donor 1	Blood of Donor 2
1 h		
NC	0	0
PC	100	100
AgNPs–SS	0.156	0.114
AgNPs–Alb	0.678	0.743
AgNPs–Tween	1.199	0.828
2 h		
NC	0	0
PC	100	100
AgNPs–SS	0.149	0.257
AgNPs–Alb	0.572	0.720
AgNPs–Tween	1.292	1.079
24 h		
NC	0	0
PC	100	100
AgNPs–SS	0.049	0.364
AgNPs–Alb	0.389	0.850
AgNPs–Tween	27.988	36.465

**Table 2 nanomaterials-11-01484-t002:** Number of MSC nuclei on the surface of the cover glasses after cocultivation with different AgNP preparations (cells/mm^2^).

Sample	Mean
Control group (nutrient medium for MSC)	1714 ± 546
Phosphate buffered saline 10 μL	1656 ± 140
Phosphate buffered saline 100 μL	1507 ± 269
AgNPs–SS 10 μL	1499 ± 278
AgNPs–SS 100 μL	1192 ± 276 *
AgNPs–Alb 10 μL	1508 ± 410
AgNPs–Alb 100 μL	1190 ± 410 *
AgNPs–Tween 10 μL	1194 ± 176 *
AgNPs–Tween 100 μL	870 ± 311 *

The data are represented as mean ± SD. * *p* < 0.001, as compared with the corresponding control group, Mann–Whitney test.

**Table 3 nanomaterials-11-01484-t003:** Cytotoxicity of different AgNP preparations when cocultivated with MSCs for 3 days.

AgNPs Preparation and Dose	Living Cells	Early Apoptosis	Necrosis/Late Apoptosis
Control	87.5 ± 3.2	3.3 ± 0.6	7.2 ± 2.6
PBS 10 µL	90.3 ± 1.3	4.7 ± 0.6	4.2 ± 1.1
PBS 100 µL	89.7 ± 1.7	4.4 ± 0.8	5.2 ± 0.5
AgNPs–SS 10 µL	68.9 ± 4.2 * ^	25.7 ± 3.2 * ^	5.3 ± 0.3
AgNPs–SS 100 µL	75.3 ± 5.0 * ^	17.2 ± 3.1 * ^	7.1 ± 0.5
AgNPs–Alb 10 µL	86.7 ± 5.3	9.5 ± 3.3 **	3.5 ± 0.5
AgNPs–Alb 100 µL	87.7 ± 0.8	7.8 ± 0.09	4.3 ± 0.6
AgNPs–Tween 10 µL	87.1 ± 0.5	8.9 ± 0.01 **	3.6 ± 0.3
AgNPs–Tween 100 µL	—	—	—

Data are expressed as mean ± SD. * *p* < 0.001 compared with control (LSD test); ** *p* < 0.05 compared with control (LSD test); ^ *p* < 0.001 compared with appropriate PBS volume.

**Table 4 nanomaterials-11-01484-t004:** Size of the growth inhibition zones after treatment of the different microbial cultures with the AgNP preparations.

AgNPs Preparation	Inhibition Zone Diameter (mm) Including the Diameter of the Well (8.0 mm)				
	Test Cultures				
	*S. aureus*	*K. pneumoniae*	*P. aeruginosa*	*Corynebacterium*	*Acinetobacter*
AgNPs–SS	* Control-0	Control-0	Control-0	Control-0	Control-0
	10 ± 1	16 ± 1	12 ± 1	16 ± 1	12 ± 1
AgNPs–Alb	Control-0	Control-0	Control-0	Control-0	Control-0
	10 ± 1	14 ± 1	12 ± 1	18 ± 1	12 ± 1

* Controls, 100 μL of sterile 0.9% NaCl was added to the wells. Data are expressed as mean ± SD, mm.

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
