# Peer review of "Hemolytic Activity, Cytotoxicity, and Antimicrobial Effects of Human Albumin- and Polysorbate-80-Coated Silver Nanoparticles"

_nanomaterials, 2021, doi:10.3390/nano11061484_

Round 1

Reviewer 1 Report

Interesting work on the analysis of the activity and properties of silver nanoparticles modified with albumin and tween.

Minor remarks:

-authors' affiliations are incomplete, no postal code, no country ...

-the corresponding author is not marked with an asterisk

-line 35- "for example" can be delated

-line 87 - "coli" in lowercase

-line 178, "In this work"  can be delated. 

-in M&M reagent catalog numbers are missing. For example, it is not known what origin was albumin? Was it human or bovine or some other? Even for bovine albumin, there are many products that differ in purity (e.g. content of other proteins, usually larger ones)

-line 280, only one value of the zeta potential is given. What was the zeta potential for each modification? For AgNPs alone, AgNPs with albumin and AgNPs with tween?

-sometimes the authors write AgNP and sometimes AgNPs in the same context. Please standardize

Author Response

May 15, 2021

Guest Editor

Nanomaterials

Dear Editors:

I, along with my coauthors, would like to thank you for consideration of our manuscript, nanomaterials-1225274, entitled “Hemolytic activity, cytotoxicity and antimicrobial effects of human albumin- and polysorbate-80-coated silver nanoparticles” for publication in the special issue of Nanomaterials as an original research article.

Please find the revised manuscript and our response to the comments by the reviewers and external editor. The changes to the text are highlighted. We hope that the present version is acceptable for publication in Nanomaterials.

Thank you for your consideration. I look forward to hearing from you.

Sincerely,

Michael Shumilo

Institute of Experimental Medicine

V.A. Almazov Federal Heart, Blood and Endocrinology Center

197341, Akkuratova Street 2, St. Petersburg, Russian Federation

E-mail: shumilo.mischa@gmail.com

Response to the comments by Reviewer 1

We would like to thank the reviewer for the fair criticism of the manuscript.

  1. Critique: “Authors' affiliations are incomplete, no postal code, no country”

Response: We have added postal codes and country.

  1. Critique: “The corresponding author is not marked with an asterisk”

Response: The corresponding author was marked with an asterisk.

  1. Critique: “Line 35- "for example" can be delated.”

Response: Line 35- "for example" has been deleted. 

  1. Critique: “Line 87 - "coli" in lowercase”

Response: Line 87 - "coli" is written in lowercase.

  1. Critique: “Line 178, "In this work"  can be delated”

Response: Line 178, "In this work" has been deleted.

  1. Critique: “In M&M reagent catalog numbers are missing. For example, it is not known what origin was albumin? Was it human or bovine or some other? Even for bovine albumin, there are many products that differ in purity (e.g. content of other proteins, usually larger ones)”

Response: We have added catalog number for human albumin, polysorbate-80 and AgNO3

  1. Critique: “Line 280, only one value of the zeta potential is given. What was the zeta potential for each modification? For AgNPs alone, AgNPs with albumin and AgNPs with tween”

Response: We have added zeta potential for each modification.

  1. Critique: “Sometimes the authors write AgNP and sometimes AgNPs in the same context. Please standardize”

Response: Thank you. We standardized designations.

Reviewer 2 Report

This manuscript reports the investigation of the cytotoxicity and antimicrobial properties of Ag NP coated with human 2 albumin or polysorbate-80. The paper the presented study is interesting and fits with the scopes of Nanomaterials, but the English needs to be improved and the manuscript should be carefully read by a person proficient in English; there are numerous typos and grammatical errors that should be corrected. Investigations on cytotoxicity of nanomaterials are important and this study investigates the effect on coating with albumin. I would recommend this manuscript for publication after revisions. Here are the comments:

  1. In the introduction, the reference about antimicrobial resistance date from 2008, the authors can keep this reference, but should add a more recent reference too.
  2. In the discussion, the authors mention green synthesis, but they did not add any references. The authors should add some references (here are possible references: DOI: 10.1016/B978-0-323-51254-1.00014-2, 10.1016/j.indcrop.2015.03.015, 10.1088/1757-899X/613/1/012003, and any other references).
  3. The authors highlight the toxicity of silver nanoparticles, but it mainly depends on the “type” of Ag NP. It is known that some Ag NP do not exhibit biocidal properties (10.1039/C5RA27836H) or antimicrobial properties(10.1007/s40089-016-0186-7). The biocidal and toxicity mechanism is not fully understood.
  4. Figure 1 is difficult to read and the figure should be improved.
  5. Figure 2 does not seem to be relevant for the study and should be replaced by XRD study.
  6. What is the interest to perform X-ray fluorescence on Ag NP? X-ray diffraction gives more information and should be performed (mandatory). The authors can estimate the average diameter of the Ag NP as the TEM image showed they are spherical.
  7. The TEM image is of very low quality and does ot meet the required criteria for being published in nanomaterials. JEM-2010 form JEOL enable to get very good high resolution images, more specifically on nanoparticles that can be easily spread on TEM grid. The TEM study should be renewed and better images should be added. A comparison between un-coated and albumin-coated Ag NP can be discussed. Does albumin induce agglomeration of Ag NP and then decrease the toxicity? The addition of TEM and XRD study will increase the interest of the reader and attract the attention of more research in the field.
  8. In figure 4, the authors should use 3 different colours for the lines in the UV-vis absorption spectra. It is difficult to discriminate the 3 different measure. The absorption peak of 2 at 590nm seems to be related to albumin. Can the authors confirm? is difficult to read and the figure should be improved.
  9. In the Table 4, the authors should add the diameter of the well (8mm). If we consider the diameter of 8mm, the inhibition zone seems quite low, more specifically for S. aureus, Acinetobacter and P. aeruginosa. Can the authors discus this low antimicrobial property? It seems also that the albumin “shield“ the antimicrobial properties.
  10. In conclusion, the authors should correct “ESCAPE” group.

Author Response

(The authors gave the same response as above.)

Reviewer 3 Report

Korolev and coauthors report silver particles with different coatings and their hemolytic and antibacterial properties. The use of albumin as a biocompatible coating with low cytotoxicity has already been studied in detail by other groups (e.g. Chanana et al.; Parak et al.; Fery et al.; etc.). The use of polysorbate-80 is also not particularly new in itself, as this coating has also been studied in detail. In summary, I find the results of this study to be of little scientific value. I can't deny that these studies also have a certain value as replication studies. In principle, I would like to recommend that the authors make greater reference to the state of the art of biocompatible coatings. And, to make it clearer to what extent the results of this study confirm or refute the already known correlations. All in all, this work could fit well into the journal Nanomaterials. However, the paper should be revised in detail. Notwithstanding, I am quite satisfied with the scientific and presentation quality.   

Comments:

  1. Please define the hemolytic activity and how it was parameterised in more detail.
  2. The authors indicate that low cytotoxity could be achieved by albumin-coating. The authors might wish to refer to reference work for protein-coated NPs e.g. by Tebbe et al. (ACS Applied Materials Interfaces) and Chanana/Parak et al.
  3. The authors discuss chemical and physical stability, however, it seems to me that they simply refer to colloidal stability. In fact chemical stability, e.g. by the oxidation/sulfoxidation or Ag was not studied.
  4. Please explain how the concentration of NPs was determined by water evaporation?
  5. Minor typos: Fig. 2 KeV should be keVFig. 4 Absorbence should be Absorbance; Fig. 4 Wavelenth should be Wavelength;
  6. Fig. 4 spectrum #3 clearly indicates an aggregation of NPs, supposedly because of poor colloidal stability. Please explain if this is the case and whether aggregation could have affected the haemolytic or antibacterial activity.
  7. In line 476 the text breaks abruptly. Please check.
  8. Section 4 (Discussion) is too detailed and very long. It might be better to summarise the core findings and to put them into the context of the state of the art.  

Author Response

May 15, 2021

Guest Editor

Nanomaterials

Dear Editors:

I, along with my coauthors, would like to thank you for consideration of our manuscript, nanomaterials-1225274, entitled “Hemolytic activity, cytotoxicity and antimicrobial effects of human albumin- and polysorbate-80-coated silver nanoparticles” for publication in the special issue of Nanomaterials as an original research article.

Please find the revised manuscript and our response to the comments by the reviewers and external editor. The changes to the text are highlighted. We hope that the present version is acceptable for publication in Nanomaterials.

Thank you for your consideration. I look forward to hearing from you.

Sincerely,

Michael Shumilo

Institute of Experimental Medicine

V.A. Almazov Federal Heart, Blood and Endocrinology Center

197341, Akkuratova Street 2, St. Petersburg, Russian Federation

E-mail: shumilo.mischa@gmail.com

Response to the comments by Reviewer 3

We would like to thank the reviewer for the fair criticism of the manuscript.

  1. Critique: “Please define the hemolytic activity and how it was parameterised in more detail”

Response: We have added relevant reference in the manuscript [Yücel, D., & Dalva, K. (1992). Effect of in Vitro Hemolysis on 25 Common Biochemical Tests. Clinical Chemistry, 38(4), 575–577. doi:10.1093/clinchem/38.4.575].

  1. Critique: “The authors indicate that low cytotoxity could be achieved by albumin-coating. The authors might wish to refer to reference work for protein-coated NPs e.g. by Tebbe et al. (ACS Applied Materials Interfaces) and Chanana/Parak et al.”

Response: We have added suggested references.

  1. Critique: “The authors discuss chemical and physical stability, however, it seems to me that they simply refer to colloidal stability. In fact chemical stability, e.g. by the oxidation/sulfoxidation or Ag was not studied.”

Response: Yes, we referred to colloidal stability only. This has been corrected in the manuscript. 

  1. Critique: “Please explain how the concentration of NPs was determined by water evaporation.”

Response: The watch glass can be used to dry solid particles and to determine the weight of the dry residue. That is, watch glass is often used in cases where a certain type of solid must be separated from its relatively volatile solvent. The substance is placed on the watch glass, and covered with filter paper to prevent contamination of the product. To maximize the drying speed, the watch glass is placed in a thermostat or fume hood to ensure good air circulation for a long time. Another method used in chemical laboratories to increase the drying rate is to pass dry air or an inert gas over the watch glass [Lehman, John W. (2008). The Student's Lab Companion: Laboratory Techniques for Organic Chemistry (2nd ed.). Prentice Hall. pp. 156–157. ISBN 9780131593817]. We have added brief description of the technique as well as the reference (lines 151-155).

  1. Critique: “Minor typos: Fig. 2 KeV should be keVFig. 4 Absorbence should be Absorbance; Fig. 4 Wavelenth should be Wavelength.”

Response: The mistakes have been corrected.

  1. Critique: “Fig. 4 spectrum #3 clearly indicates an aggregation of NPs, supposedly because of poor colloidal stability. Please explain if this is the case and whether aggregation could have affected the haemolytic or antibacterial activity.”

Response: The reviewer is probably right that spectrum #3 in Figure 4 indicates some aggregation of nanoparticles. However, the influence of this fact on the biological effects of nanoparticles coated with polysorbate-80 is, most likely, insignificant, in comparison with the chemical and toxic properties of polysorbate-80 itself as a chemical substance. We discuss the latter in the "Discussion" section. That is why, having obtained data on a significantly higher cytotoxicity and hemolytic activity of AgNPs-Tween compared to AgNPs and AgNPs-Alb, we have not used them in experiments to assess antibacterial activity.

  1. Critique: “In line 476 the text breaks abruptly. Please check.”

Response: The text has been checked.

  1. Critique: “Section 4 (Discussion) is too detailed and very long. It might be better to summarize the core findings and to put them into the context of the state of the art.”

Response: We believe that the Discussion contains important information that is essential for the reader in order to analyze the findings of the present study in the context of previous work.

Round 2

Reviewer 2 Report

The manuscript has been improved according to the suggested revisions. The novelty of this study is defined and the presented work fits the requirements for publication in Nanomaterials. I have had a careful look at the authors’ responses and at the modifications performed in the manuscript. The authors performed most of the necessary changes to improve their paper. I am satisfied of the changes made, but I would have appreciated if the authors would have added the XRD patterns. The manuscript can be accepted for publication in Nanomaterials.

Author Response

May 28, 2021

Guest Editor

Nanomaterials

Dear Editors:

I, along with my coauthors, would like to thank you for consideration of our manuscript, nanomaterials-1225274, entitled “Hemolytic activity, cytotoxicity and antimicrobial effects of human albumin- and polysorbate-80-coated silver nanoparticles” for publication in the special issue of Nanomaterials as an original research article.

Please find the revised manuscript and our response to the comments by the reviewers and external editor. We hope that the present version is acceptable for publication in Nanomaterials.

Thank you for your consideration. I look forward to hearing from you.

Sincerely,

Michael Shumilo

Institute of Experimental Medicine

V.A. Almazov Federal Heart, Blood and Endocrinology Center

197341, Akkuratova Street 2, St. Petersburg, Russian Federation

E-mail: shumilo.mischa@gmail.com

Response to the comments by Reviewer 2

Thank you for the comments above and thanks for your approval for publication. We understand your desire to see the XRD patterns, but we are still convinced that the X-ray spectra of AgNPs  has sufficient information for our research.

Reviewer 3 Report

Korolev and coauthors have provides a revised version of their contribution. I am very happy with most of the changes. This concerns mainly the improved language and the consideration of my comments. In the initial statement of my first review, my main criticisms were that the authors should (1) make greater reference to the state of the art of biocompatible coatings and (2) make it clearer to what extent the results of this study confirm or refute the already known correlations (of haemolytic activity, cytotoxicity and antimicrobial effects in regard to silver/noble metal nanocolloids). With regard to these two points of criticism, I do not find any changes in the manuscript; it remains to be assumed that the authors missed these points. As such, I would recommend that the editor allows them to submit these points in a further revision.  All in all, I rate this version as "minor revisions" since several content and technical inconsistencies (see below) need to be addressed before an acceptance of the manuscript can be recommended. As a technical note and to facilitate future revisions, it would make the review process easier if the changes regarding comments were briefly summarized in the response letter and the main text passages were cited. 

Other issues to be resolved: 

  1. In reference to Comment #8: Though the opinion of the author seems to differ from mine, I am still convinced that Section 4 (Discussion) is too detailed and even wordy at time. I fails to put the results and findings in context to the state-of-the-art. I would prefer if the authors would make this section more concise and would present a brief conclusion section with the main conclusions for the sake of clarity. 
  2. Figures 1 and 4 use decimal comma instead of points. I discourage the use of two different decimal notations. Please use either comma or point throughout the manuscript.
  3. Figure 2 suffers from low resolution. Please revise.
  4. Figure 3 seems messy owing to different image sizes mixed randomly. I would prefer to have the thee images neatly next to each other, all of the same dimensions. Please revise.
  5. The scale bars in Figure 5 are too small to be legible. Please revise. 

Author Response

May 28, 2021

Guest Editor

Nanomaterials

Dear Editors:

I, along with my coauthors, would like to thank you for consideration of our manuscript, nanomaterials-1225274, entitled “Hemolytic activity, cytotoxicity and antimicrobial effects of human albumin- and polysorbate-80-coated silver nanoparticles” for publication in the special issue of Nanomaterials as an original research article.

Please find the revised manuscript and our response to the comments by the reviewers and external editor. We hope that the present version is acceptable for publication in Nanomaterials.

Thank you for your consideration. I look forward to hearing from you.

Sincerely,

Michael Shumilo

Institute of Experimental Medicine

V.A. Almazov Federal Heart, Blood and Endocrinology Center

197341, Akkuratova Street 2, St. Petersburg, Russian Federation

E-mail: shumilo.mischa@gmail.com

Response to the comments by Reviewer 3

  1. Issue: “In reference to Comment #8: Though the opinion of the author seems to differ from mine, I am still convinced that Section 4 (Discussion) is too detailed and even wordy at time. I fails to put the results and findings in context to the state-of-the-art. I would prefer if the authors would make this section more concise and would present a brief conclusion section with the main conclusions for the sake of clarity.”

Response: We have revised and shortened the discussion, and have highlighted the сonclusions in a separate section.

  1. Issue: “Figures 1 and 4 use decimal comma instead of points. I discourage the use of two different decimal notations. Please use either comma or point throughout the manuscript.”

Response: In Figures 1 and 4 we have replaced commas with points.

  1. Issue: “Figure 2 suffers from low resolution. Please revise.”

Response: We put the pictures next to each other and made them the same size. 

  1. Issue: “Figure 3 seems messy owing to different image sizes mixed randomly. I would prefer to have the three images neatly next to each other, all of the same dimensions. Please revise.”

Response: We put the pictures next to each other and made them the same size.

  1. Issue: “The scale bars in Figure 5 are too small to be legible. Please revise.”

Response: We increased the size of the picture so that you can see the scale bars.
